# Data Mining Suggests That CXCL14 Gene Silencing in Colon Cancer Is Due to Promoter Methylation

**DOI:** 10.3390/ijms242216027

**Published:** 2023-11-07

**Authors:** Yanjing Wang, Siyi Wang, Yuchen Niu, Buyong Ma, Jingjing Li

**Affiliations:** Engineering Research Center of Cell & Therapeutic Antibody, School of Pharmacy, Shanghai Jiao Tong University, Shanghai 200240, China; wangyanjing@sjtu.edu.cn (Y.W.); wangsiyi_sjtu@163.com (S.W.); nyc123@sjtu.edu.cn (Y.N.)

**Keywords:** colorectal cancer, CXCL14, gene silence, metastasis, DNA methylation

## Abstract

CXCL14 is one of the most evolutionarily conserved members of the chemokine family and is constitutionally expressed in multiple organs, suggesting that it is involved in the homeostasis maintenance of the system. CXCL14 is highly expressed in colon epithelial cells and shows obvious gene silencing in clinical colon cancer samples, suggesting that its silencing is related to the immune escape of cancer cells. In this paper, we analyzed the expression profiles of multiple human clinical colon cancer datasets and mouse colon cancer models to reveal the variation trend of CXCL14 expression during colitis, colon polyps, primary colon cancer, and liver metastases. The relationship between CXCL14 gene silencing and promoter hypermethylation was revealed through the colorectal carcinoma methylation database. The results suggest that CXCL14 is a tumor suppressor gene in colorectal carcinoma which is activated first and then silenced during the process of tumor occurrence and deterioration. Promoter hypermethylation is the main cause of CXCL14 silencing. The methylation level of CXCL14 is correlated with the anatomic site of tumor occurrence, positively correlated with patient age, and associated with prognosis. Reversing the hypermethylation of CXCL14 may be an epigenetic therapy for colon cancer.

## 1. Introduction

Chemokines are cytokines that appeared with the emergence of chordates. There are 41 ligands and 21 receptor genes in the human genome [1]. The chemotactic action of chemokines induces the effector cells to move against the gradient through the concentration gradient formed by the local tissue expression and secretion. This system plays an important role in embryonic development and immune cell homing. In addition to initial chemotaxis, chemokines are also involved in metabolic regulation [2], angiogenesis [3], and tissue development [4].

CXCL14 is one of the earliest members to appear in the chemokine family [5]. Compared with other members, it evolves extremely slowly and nonsense SNPs are rare, indicating its significant function. In healthy people, CXCL14 is usually highly expressed in tissues with a rapid renewal rate, such as epithelial cells of the small intestine, colorectal, gastric mucosa, endometrium, mammary gland, skin, and other tissues. It is low in resting state cells, such as the liver, muscle, connective tissue, brain, and other parenchymal cells. CXCL14 has shown abnormal expression levels in some immune diseases and a large number of tumor samples [6,7,8]. CXCL14 gene knockout has effects on the survival rate and metabolism of sugar and lipids in mouse embryos [9].

Contrary to its important function, the receptor for CXCL14 has not yet been discovered. Kouzeli et al. [10] systematically studied the interaction between CXCL14 and all genes of the chemokine receptor family and found that CXCL14 could not activate anyone. However, it is interesting that CXCL14 can synergistically promote the physiological activity of several other chemokines. Hundelshausen et al. [11] systematically studied the pattern of heterodimers formed by chemokines and found that a large number of heterodimers existed in the chemokine family. It is suggested that CXCL14 functions by promoting or inhibiting the activity of other chemokines. This may explain why the effects of CXCL14 may be opposite in different tissues and tumor types [12] because the chemokines that partner with CXCL14 in different tissues can vary greatly in both type and concentration. CXCL14 appears as a tumor suppressor in colon cancer.

Tumor development is the result of the activation of oncogenes and/or inactivation of tumor suppressor genes. It was previously thought that these genetic variations were all due to genetic mutations [13]. However, it is now known that epigenetic variation is another pathway that leads to the silencing of tumor suppressor genes and that abnormal expression of genes caused by abnormal DNA methylation is more frequent than changes caused by gene copy number [14]. We investigated the methylation level variation and gene mutation frequency of colon cancer tumors and found that the CXCL14 promoter was particularly hypermethylated in colon cancer samples.

The relationship between CXCL14 hypermethylation and gene silencing was also found in some tumor types. Tessema et al. found that CXCL14 was epigenetically silenced in lung cancer and considered this gene to be an important epigenetic therapeutic site [7]. Cao et al. observed that the epigenetic silencing of CXCL14 in colon cancer is a mechanism leading to tumor metastasis and invasion [15]. Hu et al. also found that abnormal methylation of promoters in gastric tumors resulted in inhibition of CXCL14 expression [16].

To further reveal the changes of CXCL14 in colon cancer, this paper studied the changes of CXCL14 expression from normal tissue to colitis, to the primary tumor, and eventually to metastatic cancer by mining multiple datasets. We explored the changes of CXCL14 methylation in colon cancer and its correlation with CXCL14 gene silencing, prognosis, and important clinical markers and revealed the silencing mechanism of CXCL14 and its important significance in the development of colon cancer.

## 2. Result

### 2.1. CXCL14 Silences as the Colorectal Cancer Progresses

We mined four groups of human colon cancer expression profile data sets and found that CXCL14 expression levels had obvious changes in different stages of colon cancer development. Figure 1A shows RNA microarray data from 259 colon cancer samples and 12 human cell lines collected at Memorial Sloan Kettering Cancer Center between 1992 and 2004 [17] (Appendix A). From the data, we can clearly see the expression changes of CXCL14 in different cancer stages. Firstly, CXCL14 expression levels were normally distributed in healthy tissues but after the occurrence of intestinal polyps, the expression level dispersion increased, indicating a slight dysregulation in the expression level. After further development into primary tumors, the dispersion of expression levels increased further and the median decreased, indicating that the dysregulation of the expression level was intensified and the selection of cancer cells in vivo was progressing in the direction of CXCL14 silencing. The phenomenon of gene silencing became more significant after liver metastasis or lung metastasis, indicating that CXCL14 silencing has certain advantages for tumor metastasis or tumor formation after metastasis. CXCL14 silencing occurred completely in 12 tumor cell lines.

In the dataset E-GEOD-4183 (Figure 1B), Galamb et al. collected 15 patients with CRC, 15 with adenoma, and 8 healthy normal controls [18]. It was also found that in inflammatory bowel disease, although the mean expression of CXCL14 did not change significantly, the dispersion increased and the CV% increased from 24% in healthy tissues to 83% in colitis tissues, indicating dysregulation of CXCL14 expression. CXCL14 levels decreased significantly with adenoma and colorectal cancer (CRC) (*p* < 0.001).

In another set of expression profile data of colon cancer and adjacent tissues (Figure 1C), it was also found that the expression of CXCL14 in colon cancer tissues decreased and was accompanied by an increase in dispersion.

In the last set of data, which was about a clinical follow-up investigation, Martineau et al. tracked and collected primary cancer tissue, metastatic tissue, and healthy colon tissues from 19 tumor patients and performed RNA microarray analysis [19]. We analyzed the expression of CXCL14 and found that CXCL14 expression continued to decline during tumor progression (Figure 1D–G, Appendix A).

From the above four sets of expression profile data, it can be seen that CXCL14 silencing becomes more and more significant with the increase in colon cancer malignancy.

After that, we investigated the expression profiles of two independent mouse colon cancer models. In the first experiment (Figure 1H), two modeling methods were used to establish the primary model of colon cancer in mice [20]. In the first, the CAC model was induced by AOM/DSS and the second was spontaneous colon cancer modeling by C57BL/6-ApcMinC/Nju transgenic mice (SporCRC). The expression profiles of cancer tissues and adjacent tissues in the CAC group and SporCRC group were detected and it was found that the CXCL14 expression level in tumor tissues was higher than that in corresponding healthy control tissues. In the second model (Figure 1I) [21], the authors used a single AOM/DSS model and it was observed that the expression level of CXCL14 increased.

Therefore, the expression changes of CXCL14 in human colon cancer and in mouse colon cancer development seem to be opposite; whether they are contradictory will be discussed in the following section.

Clinical data show that patients with higher CXCL14 expression have a better prognosis. Two colon cancer datasets from two databases, the Protein Atlas (Figure 2A, Appendix A) and Gepia (Figure 2B), showed that higher CXCL14 expression was associated with longer survival. In the newly published colon cancer dataset (Sidra-LUMC AC-ICAM, Nat Med 2023) [22], the CXCL14 mRNA level is positively correlated with overall survival (OS) time (Figure 2C, Appendix A).

### 2.2. CXCL14 Silencing Is Associated with Promoter CpG Methylation

CXCL14 rarely has gene mutations in colon cancer. We analyzed the colorectal adenocarcinoma dataset (TCGA, PanCancer Atlas) and found that in clinical samples of colon cancer, only 0.6% of total 526 samples had genetic variation, in which 1 sample had gene deletion and 2 samples had nonsense mutations.

We then analyzed the colon cancer methylation dataset [23] and found that CXCL14 methylation in colon cancer tissues showed significant hypermethylation across the entire promoter region as well as the 5′ untranslated region. Especially in the range −103~90 (probe 6, 7, 8), the hypermethylation was extremely significant (Figure 3A). A comparison of cancer/paracancer tissue pairs showed that methylation levels in cancer tissue were significantly higher in more than 75% of samples than in paracancer control tissue (a more than three-fold increase, Figure 3C–E). At probes 6, 7, and 8, samples were divided into hypermethylation and hypomethylation groups; the student *t*-test result denoted that the mRNA level in the two groups is significantly different (Figure 3F,G). In the following analysis, the average β-value of probes 6, 7, and 8 in each sample was used.

### 2.3. The Methylation of CXCL14 in Colon Cancer Is Closely Related to the Anatomical Site and Patient Age

We further analyzed the association between CXCL14 hypermethylation and clinical parameters (Table 1, Appendix A). Based on the upper limit of the methylation level of CXCL14 in healthy tissues 0.1, cancer tissues were divided into the CXCL14 unmethylated group (β-value < 0.1) and the hypermethylated group (β-value > 0.1). A comparison between the non-methylated group and the hypermethylated group showed that CXCL14 hypermethylation was associated with the anatomical site (*p* < 10^−6^), age at diagnosis (*p* < 0.0001), and the clinical stage of the tumor (*p* = 0.019). Comparing the tumor cells from different colonic sites, it was found that the methylation level of the tumor cells originating from the anterior segment of the colon (cecum, ascending colon, hepatic flexure, and transverse colon) was significantly higher than that of the control paracancer tissues. However, the methylation of tumor cells from the posterior segment of the colon (descending colon and sigmoid colon) was significantly lower than that of the first half. Accordingly, CXCL14 expression decreased significantly in the anterior segment of the colon compared to the posterior segment (Figure 4B).

### 2.4. CXCL14 Methylation and Senescence

By correlation analysis, it was found that CXCL14 methylation increased slowly with age in normal colon tissue and that the correlation was very strong (r = 0.58, *p* < 0.0001). However, the methylation trend of CXCL14 in tumor samples showed a distinct differentiation. We used the upper limit of CXCL14 methylation level in normal colon samples, β-value = 0.1, as the boundary to divide tumor samples into hypermethylated and hypomethylated groups. The hypermethylated group accounted for 72% of the total samples. Although elevated methylation was still linearly correlated with age in this sample, the degree of correlation was significantly reduced (r = 0.194, *p* = 0.0047). On the contrary, in the hypomethylated group (28% of total tumor samples), the methylation accumulation process tended to stall and did not increase with age, which was a different mode from both normal tissue and hypermethylated tumor samples.

### 2.5. CXCL14 Methylation Was Associated with Survival

We analyzed the correlation between methylation levels and survival at different probe sites and found that higher methylation level at probes 3, 9, 11, and 12 was associated with worse prognosis while high methylation levels at probe 6, 7, and 8, in contrast, was associated with longer survival (Figure 5D,E). Since we had observed abnormally low methylation in some cases in Figure 4D, we excluded these cases and reanalyzed the survival time and found that high methylation level was associated with poor prognosis (Figure 5I,J), although the correlation was not significant.

### 2.6. The Methylation and Demethylation of the CXCL14 Promoter Directly Regulate CXCL14 Expression

We treated human colon cancer cells, HT29 and HCT116, with DAC and found that CXCL14 expression was positively correlated with the DAC dose and treatment time (Figure 6A–C). We cloned the sequence of the human CXCL14 promoter from upstream −1000 bp to downstream 100 bp from the transcription start site (TSS) and inserted the upstream of pGL4.11 plasmid Luc2p gene. Then, through promoter truncation, key transcription factor binding sites were found located in the −170 bp~100 bp region (Figure 6D). After the −170~100 region was methylated with M. SSSI enzyme, the promoter activity was significantly inhibited, indicating that the methylation of this region has an inhibitory effect on CXCL14 promoter activity (Figure 6E).

## 3. Methods

### 3.1. mRNA Expression Data

The four groups of human colon cancer expression profiles analyzed in this study are as follows. (1) Memorial Sloan Kettering Cancer Center, a total of 390 microarray data (E-GEOD-41258) including primary colon adenocarcinomas, adenomas, metastasis, and corresponding normal mucosae [17]. (2) Different stage colorectal carcinomas (CRC) and inflammatory bowel diseases (IBD) (E-GEOD-4183), including biopsies of 15 patients with CRC, 15 with adenoma, 15 with IBD, and 8 healthy normal controls [18]. (3) In total, 36 CRC tissues and 24 non-cancerous colorectal tissue. (E-GEOD-23878). (4) Pairs of primary tumors, hepatic metastases, and normal controls [19].

Two mouse primary colorectal cancer microarray datasets were analyzed. They are about AOM/DSS induced colitis-associated cancer (CAC) model [20,21] and ApcMin/+/J transgenic mice-based spontaneous tumor model [21].

### 3.2. DNA Methylation Data

We used the mexpress database (http://mexpress.be, accessed on 3 November 2022) to study the methylation status of the CXCL14 gene in the TCGA-COAD dataset [23]. The methylation of sites of probes 6, 7, and 8, that were significantly hypermethylated in clinical colorectal carcinoma were used to analyze associations with clinical indexes in Table 2.

### 3.3. Survival Analysis

Cases were separated into groups of higher or lower CXCL14 mRNA expression by the best expression cut-off of the default setting of the Human Protein Atlas (https://www.proteinatlas.org/, accessed on 14 May 2023) and Gepia2 (http://gepia2.cancer-pku.cn, accessed on 14 May 2023) platforms. For survival analysis of CXCL14 methylation, the cases were split into groups of hypermethylation and hypomethylation by the maxstat algorithm. Overall survival curves were achieved using the Kaplan–Meier assessment and the log-rank test.

### 3.4. Cell Culture

293T (ATCC and CRL-3216) and HT29 (ATCC and HTB-38) cells were cultured in RPMI1640 medium containing 10% FBS and HCT116 (ATCC, CCL-247) cells were cultured in McCoy’s 5A containing 10% FBS. The cell culture conditions were 37 °C, 5% carbon dioxide, and 70% humidity. Decitabine (DAC) was purchased from MCE (HY-A0004, Shanghai, China) and formulated with DMSO for 10 mM storage. When DAC is treated with cells, DAC is diluted to the desired concentration in the medium and added to the cell culture dish.

### 3.5. Real-Time Quantitative PCR

Cell total RNA was extracted using the TaKaRa MiniBEST Universal RNA Extraction Kit (Cat 9767, Takara, Shiga, Japan). Reverse transcription was performed using the HiScript II 1st Strand cDNA Synthesis Kit (Cat R212-01, Vazyme, Nanjing, China). Quantitative PCR was performed using the TAKARA SybrGreen Kit (RR820Q, Takara, Shiga, Japan). The quantitative PCR primer for human CXCL14 was 5′-AAGCCAAAGTACCCGCACTG (forward), 5′-GACCTCGGTACCTGGACACG (reverse); the reference gene β-actin primers are 5′-GTGAAGGTGACAGCAGTCGGTT (forward) and 5′-GAAGTGGGGTGGCTTTTAGGAT (reverse). The relative expression of target genes was calculated by the ΔΔCt method.

### 3.6. Promoter Assay

The plasmid pGL4.11 [Luc2p] was used as vector for the promoter assay in which truncated CXCL14 promoter fragments were cloned between KpnI and HindIII sites (Figure 6D). When studying the effect of methylation on promoters, CpG methyltransferase M. SSSI (EM0821, Thermo Scientific, Shanghai, China) was used to methylate PCR products and the PCR recovery kit was used to recover reaction products. The methylated fragments were then ligated into the vector using the Gibson Assembly Cloning Kit (C115, Vezyme, Nanjing, China) and the product concentration was measured by spectrophotometry. The 293T cells in the 24-well plate were transfected with 0.9 µg of promoter plasmid and 0.1 µg of the pEGFP-N1 plasmid using Lipofectamin 2000 (Invitrogen, Carlsbad, CA, USA). After 48 h, the cells were digested with trypsin and loaded into a white wall transparent bottom 96-well plate. The green fluorescence (EGFP) was measured with a Tecan M200 pro microplate (Männedorf, Switzerland) reader and the chemiluminescence intensity was then measured immediately after adding 150 μg/mL D-luciferin. The Luc/EGFP value was used as the promoter activity.

### 3.7. Statistical Analysis

SPSS software v25 was used for statistical analysis in this study. The student *T*-test was used to assess the significance between two sets of data. Paired *T*-tests were used to test the significance of differences between paired data from the same individual. Chi-square tests were used to examine the association between CXCL14 methylation and different clinical indexes. Spearman’s correlation approach was examined in methylation β-value and patient ages. *: *p* < 0.05, **: *p* < 0.01, ***: *p* < 0.001.

## 4. Discussion

The popularization of high-throughput sequencing technology has provided a powerful tool for cancer research and generated a large number of tumor datasets with important values. Through the analysis of these data, the role of CXCL14 in colon cancer progression has been paid more and more attention by researchers. Zhou et al. revealed the regulatory role of CXCL14 in tumor metastasis and tumor immune cell composition through the single cell sequencing data set of tumor tissues [24]. Liu et al. revealed the inhibitory effect of CXCL14 on liver metastasis of colon cancer by analyzing the mRNA dataset of colon cancer [25].

In this study, we found significant differences in the regulation of CXCL14 expression between human colon cancer tumors and mouse colon cancer models. CXCL14 is generally silenced in clinical tumor tissues while it is upregulated in primary mouse tumors. Whether this contradiction is based on the inherent differences between humans and mice or some other reason deserves further discussion. First, we know that human tumor tissues progress for years to decades before being discovered so the clinical tumor samples we observe are the result of many mutations and clonal differentiation and, at the same time, many years of selection in vivo. Whereas, the mouse tumor samples were all primary tumors that developed within a few months with limited time for mutation and selection. So, there is a difference in the stage between the human and mouse tumors. Secondly, it should be noted that although the expression of CXCL14 decreased with the increase in cancer cell malignancy in a general trend, the expression of CXCL14 increased in some tumors (Figure 1A–D).

Taking the above two points together, we propose that the up-regulation of CXCL14 in the primary tumor of mice is not inconsistent with its silencing in clinical samples but with its expression at different stages of tumor development. The mouse tumor model revealed the gene regulation pattern of CXCL14 in the early stage of tumor formation. As a result of the fact that there are some currently unknown mechanisms, CXCL14 is activated during tumor formation, possibly by recruiting immune cells to participate in the tumor immune response. The clinical tumor samples reflect the CXCL14 status in the middle and late stages of tumor development. It is the result of genetic and/or epigenetic variation and has undergone long-term evolution and subclonal selection. However, this hypothesis is difficult to confirm in clinical samples because early latent tumors are difficult to obtain clinically so it is difficult to prove the hypothesis that CXCL14 is activated and then silenced during tumorigenesis. This hypothesis may be proved by long-term systematic animal tumor research.

The mechanism of CXCL14 inhibiting the development and metastasis of colon cancer may come from various aspects. Firstly, CXCL14 can directly inhibit the proliferation, migration, and invasion of tumor cells [15,26]. Secondly, this gene can regulate the abundance of immune cells in the tumor microenvironment [27,28]. Finally, CXCL14 is also an angiogenic inhibitor. Shellenberger et al. first studied the inhibitory effect of CXCL14 on angiogenesis in mouse cornea [3]; Liu et al. uncovered that CXCL14 could inhibit angiogenesis in liver cancer [29]; and Zhou et al. revealed through database mining that CXCL14 had a negative regulatory effect on angiogenesis in colon cancer, although not statistically significant [24]. Although there is no direct evidence that CXCL14 inhibits colon cancer angiogenesis, it is a question that deserves further investigation.

The silencing of CXCL14 in colon cancer is mainly due to promoter hypermethylation. Since the database shows that the mutation rate of CXCL14 gene is very low in various tumor samples (not shown), the gene mutation is not the cause of CXCL14 silencing. Why, in the tumor genome, are some genes are more prone to change their expression levels through gene mutations, such as the well-known p53, while others, such as CXCL14, are more prone to epigenetic silencing? We believe that compared with the simple deletion of genes, gene missense mutation can produce new phenotypes, one of the important aspects of which is to promote tumor metastasis. For example, some p53 mutants not only lose the cancer-suppressing function of the wild-type gene but also interfere with other signaling pathways, thus obtaining survival advantages through the gain-of-function mechanism [30]. With chemokines like CXCL14, tumor cells could gain a survival advantage by simply reducing immune surveillance through gene silencing. Due to the fact that epigenetic modification is less stable than genetic information, in the process of tumor progression, epigenetic silencing becomes the fate of the genes that are more sensitive to epigenetic regulation in tumor cells.

CXCL14 methylation levels were correlated with tumor anatomical location. Firstly, we report that the level of CXCL14 promoter methylation in colon cancer is correlated with the anatomical site of the primary tumor. The data showed that tumors from the anterior segment of the colon had a greater methylation increase. Although the methylation level from the posterior segment of colon cancer is still elevated compared with normal paracancer tissue, the extent of the methylation level increase is significantly decreased compared with the tumor from the anterior segment. This result is interesting but we cannot yet explain why this phenomenon occurs.

We also found a relationship between the degree of CXCL14 methylation and age. In healthy colon tissue, CXCL14 methylation was significantly positively correlated with age (r = 0.58); indeed, the regression line almost passes through the origin of coordinates. This fits well with the epigenetic clock model [31]. For tumor tissue, the samples can be divided into two subgroups: one group showed hypermethylation while the other one showed hypomethylation. Hypermethylated groups exhibit faster methylation accumulation than healthy tissues, which is consistent with what has been reported in the literature [32]. However, in the hypomethylated group, the epigenetic clock stopped completely and methylation levels did not accumulate with age. Whether this group of tumors shows a reversal of epigenetic aging and which kind of tumors have better prognosis compared with those that accelerate epigenetic aging is a question worthy of attention.

We investigated the correlation between methylation level and survival and found that methylation level was significantly negatively correlated with overall survival at probe positions 3–5 and 9–12 but was opposite at probe positions 6, 7, and 8 (Figure 5C–E). To explain this contradiction, we specifically excluded the abnormal hypomethylated samples at 6, 7, and 8 probe locations and re-performed correlation analysis. We found that after such treatment, methylation levels at these three probe locations were negatively correlated with survival, which was consistent with the trend in the other locations. Therefore, we believe that the hypomethylated cases at the 6, 7, and 8 probe location show different characteristics from the hypermethylated cases; they have a worse prognosis than the CXCL14 promoter hypermethylated cases and the mechanism needs to be further studied.

The truncated promoter assay revealed that −170~0 bp from TSS has a significant promoter activity which is coincident with the CpG island position of the promoter. Here, we used an in vitro methylation experiment which proved that the activity of this region is sensitive to CpG methylation.

In summary, this paper describes the dynamic profile of CXCL14 expression with tumor development from the time axis through bioinformatics mining. The results revealed a dynamic process in which CXCL14 expression is up-regulated in early cancerous tissues and goes to silencing in metastatic tumors, suggesting that colon cancer cells gain proliferation and metastasis advantages through CXCL14 silencing caused by its promoter methylation. Survival analysis confirmed that the gene silencing of CXCL14 in clinical colon cancer samples was associated with poor prognosis and revealed its anti-colon cancer function. In vitro experiments have shown that CXCL14 promoter methylation is an important cause of gene silencing. However, the mechanism of CXCL14 silencing promoting the progression of colon cancer is still unclear which may be related to the immune surveillance of CXCL14 [3,27,28,33] or/and antiangiogenic function [29].

## Figures and Tables

**Figure 1 ijms-24-16027-f001:**
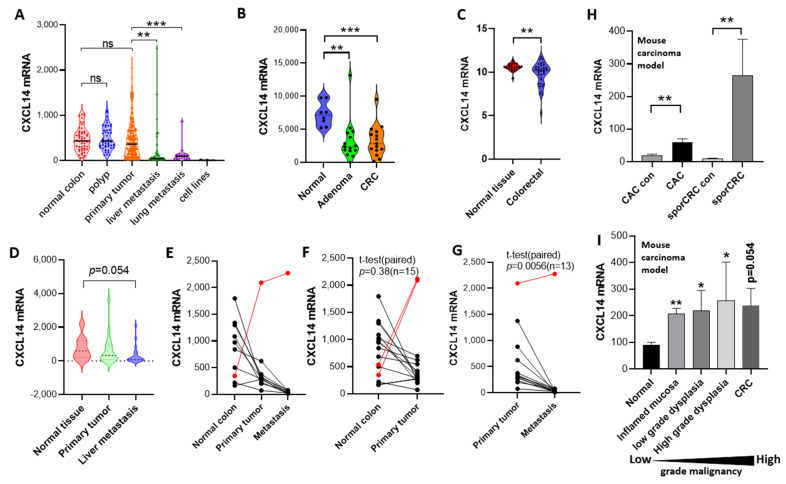
CXCL14 was downregulated in clinical colorectal carcinoma. (**A**) CXCL14 expression level in 259 samples of different stages of tumorigenesis and 10 colorectal carcinoma cell lines. (**B**) CXCL14 expression level of E-GEOD-4183 dataset, including 15 patients with CRC, 15 with adenoma, 15 with IBD, and 8 healthy normal controls. (**C**) CXCL14 expression profile in E-GEOD-23878 dataset with 35 CRC and 24 normal samples. (**D**) Overview of CXCL14 expression in E-GEOD-49355 dataset of normal colon, primary tumor, and liver metastasis samples of 13 patients. (**E**) Line chart of CXCL14 expression of different sample types. (**F**,**G**) Paired comparisons of normal primary tumor and primary metastasis tumor. The red lines highlight the samples that go against the gene silence trend. (**H**) Expression profile in the mouse primary colon cancer model. (**I**) Expression profile in the AOM/DSS-induced CRC model. *: *p* < 0.05, **: *p* < 0.01, ***: *p* < 0.001. ns: not significant.

**Figure 2 ijms-24-16027-f002:**
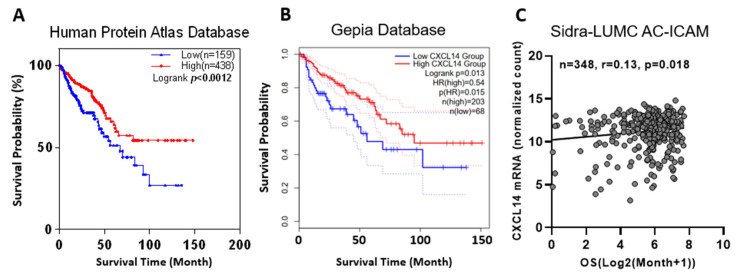
Survival curves of colon cancer with different CXCL14 expression levels. (**A**) Kaplan–Meier survival curve of TCGA database Colon Adenocarcinoma (TCGA-COAD). (**B**) Survival curve of the Gepia COAD dataset. (**C**) Correlation of overall survival (OS) of the Sidra-LUMC AC-ICAM dataset.

**Figure 3 ijms-24-16027-f003:**
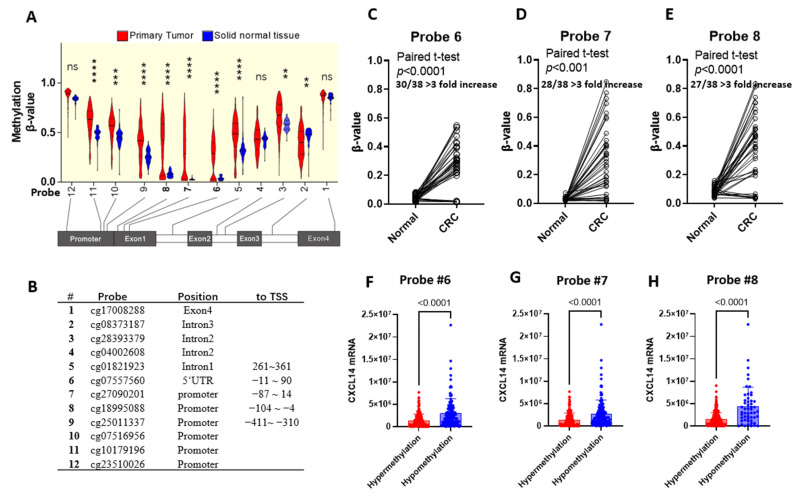
CXCL14 promoter was hypermethylated in the primary tumor. (**A**) Tumor sample and normal control of the TCGA dataset; 5 mC was sequenced by bisulfite sequencing. (**B**) Probe number in (**A**). (**C**–**E**) Paired samples of the primary tumor and normal control were compared in the #6~8 locus. (**F**–**H**) The correlation between RNA expression and methylation level was assayed by a correlation study. **: *p* < 0.01, ***: *p* < 0.001, ****: *p* <0.0001. ns: not significant.

**Figure 4 ijms-24-16027-f004:**
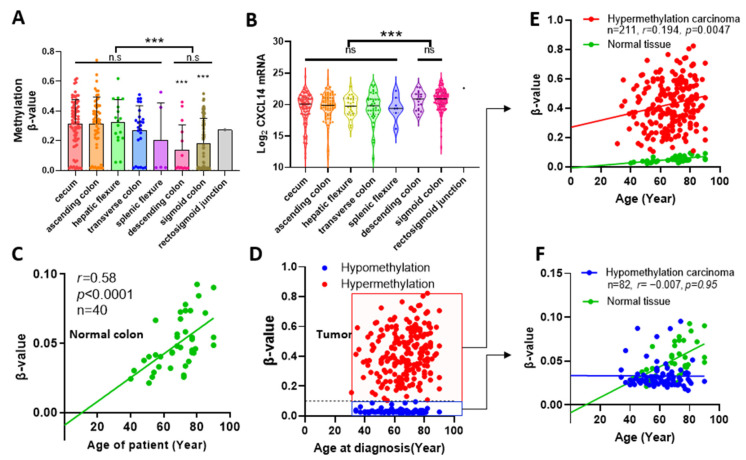
Methylation of the CXCL14 promoter was correlated with the anatomic neoplasm subdivision of the primary tumor and age at diagnosis. (**A**) The CXCL14 promoter methylation level in the different anatomic subdivisions of the colon. (**B**) CXCL14 mRNA level in different anatomic subdivisions of the colon. (**C**) The correlation analysis of the CXCL14 methylation level of normal colon tissue and donor’s age. (**D**) Tumor samples were separated into hyper- and hypo-methylation groups by 0.1 of β-value. (**E**) The group of hypermethylation showed a mild correlation between β-value and age at diagnosis. (**F**) There is no correlation in the group of hypomethylation. *** *p* < 0.001. ns: not significant.

**Figure 5 ijms-24-16027-f005:**
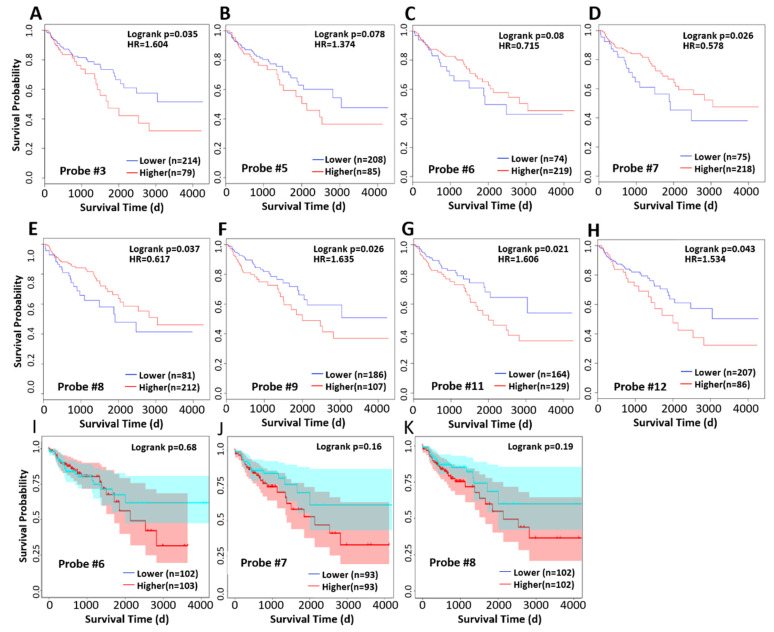
Methylation of the CXCL14 promoter is associated with prognosis. (**A**–**H**) Kaplan–Meier survival curve of higher and lower CXCL14 promoter methylation group for different probes in the TCGA colorectal database. (**I**–**K**) Kaplan–Meier survival curve examination of overall survival in subjects with high and low methylation levels in the hypermethylation group, as mentioned in Figure 4D.

**Figure 6 ijms-24-16027-f006:**
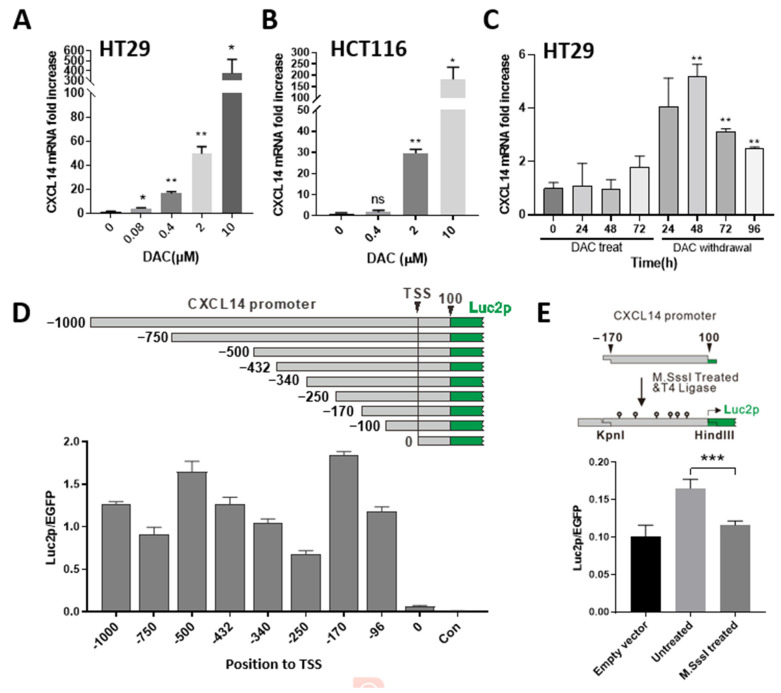
CXCL14 promoter methylation is negatively correlated with transcription level. (**A**) HT-29 and (**B**) HCT116 were treated with DAC with gradient concentration and CXCL14 silencing was reversed. (**C**) CXCL14 transcription in HT29 was activated by DAC treatment and decreased after drug withdrawal. (**D**) Plasmids containing different lengths of truncated CXCL14 promoter were evaluated by a luciferase assay. (**E**) Methylation of the −170~100 region of CXCL14 promoter by M.SssI treatment decreases CXCL14 expression. *: *p* < 0.05, **: *p* < 0.01, ***: *p* < 0.001.

**Table 1 ijms-24-16027-t001:** Dataset used in the study.

	Species	Sample Type	Sample Number	Accession No.	Reference
1	Human	Primary colon adenocarcinomas, adenomas, metastasis, and corresponding normal mucosae	390	E-GEOD-41258	[17]
2	Human	15 CRC, 15 with adenoma, and 8 healthy normal controls	38	E-GEOD-4183	[18]
3	Human	36 CRC tissues and 24 non-cancerous colorectal tissue	60	E-GEOD-23878	
4	Human	Paired primary tumors and hepatic metastases and normal controls	57	E-GEOD-49355	[19]
5	Human	Colorectal adenomas, adenocarcinomas, and normal control	336 ^†^/556	TCGA-COAD PanCancer Atlas	[23]
6	Mouse	AOM/DSS induced CAC, ApcMin/+/J spontaneous tumor and para-carcinoma tissue	14	E-GEOD-43338	[20]
7	Mouse	Ulcerative CAC	18	E-GEOD-31106	[21]

^†^: Samples with methylation data.

**Table 2 ijms-24-16027-t002:** Correlation of CXCL14 methylation and clinical indexes.

Clinical Parameters		Number		CXCL14 Methylation Status				
				Unmethylated n = 84 (28%)		Methylated n = 212 (72%)		*p*-Value (χ^2^)
**Gender**	Female	136	(55%)	36	(44%)	100	(47%)	0.614
	Male	112	(45%)	46	(56%)	112	(53%)	
**Race**	Asian	12	(4%)	3	(4%)	9	(5%)	0.33
	Black	58	(21%)	21	(27%)	37	(19%)	
	White	206	(75%)	54	(69%)	152	(77%)	
**Age**	<60	98	(33%)	43	(52%)	55	(26%)	<0.0001
	≥60	196	(67%)	39	(48%)	157	(74%)	
**anatomic** **neoplasm** **subdivision**	cecum	75	(27%)	13	(16%)	62	(31%)	<10^6^
	ascending colon	55	(20%)	11	(14%)	44	(22%)	
	hepatic flexure	22	(8%)	0	(0%)	22	(11%)	
	transverse colon	25	(9%)	6	(8%)	19	(10%)	
	splenic flexure	5	(2%)	3	(4%)	2	(1%)	
	descending colon	14	(5%)	9	(11%)	5	(3%)	
	sigmoid colon	83	(30%)	38	(48%)	45	(23%)	
**Poly**	Yes	77	(35%)	24	(39%)	53	(34%)	0.457
	No	140	(65%)	37	(61%)	103	(66%)	
**Kras gene** **mutation**	Yes	21	(48%)	11	(61%)	10	(38%)	0.139
	No	23	(52%)	7	(39%)	16	(62%)	
**lymphatic** **invasion**	Yes	79	(31%)	25	(36%)	54	(29%)	0.268
	No	180	(69%)	45	(64%)	135	(71%)	
**microsatellite** **instability**	Yes	11	(13%)	10	(17%)	1	(3%)	0.126
	No	77	(87%)	48	(82%)	29	(96%)	
**New tumor** **event**	Yes	62	(23%)	17	(23%)	45	(23%)	0.998
	No	208	(77%)	57	(77%)	151	(77%)	
**Perineural** **invasion** **present**	Yes	45	(25%)	14	(30%)	31	(24%)	0.423
	No	132	(75%)	33	(70%)	99	(76%)	
**neoplasm** **cancer status**	Yes	62	(25%)	23	(33%)	39	(22%)	0.069
	No	187	(75%)	47	(67%)	140	(78%)	
**Tumor** **stage**	I	44	(15%)	6	(8%)	38	(18%)	0.019
	II	114	(40%)	30	(39%)	84	(41%)	
	III	85	(30%)	23	(30%)	62	(30%)	
	IV	41	(14%)	18	(23%)	23	(11%)	

## Data Availability

No new data were created.

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
