# Peer review of "Data Mining Suggests That CXCL14 Gene Silencing in Colon Cancer Is Due to Promoter Methylation"

_ijms, 2023, doi:10.3390/ijms242216027_

Round 1

Reviewer 1 Report

Comments and Suggestions for Authors

Comment 1:  In 2023, a very similar study was published by Lei Zhou and published in the BMC Gastroenterol. In my opinion, an update of the literature should take into consideration this study, adding what has been published in the last two years.

Comment 2: Have the authors cross-checked their results about CXCL14  in the available Colon cancer data in cBioPortal.

Comment 3: The authors mentioned in the methodology section about HCT15 cell lines, but not shown in the results section, can authors discuss the results of this HCT15 cell lines?

Comment 4:  Could the authors add more in the conclusion about the role of CXCL14 in the progression of colon cancer angiogenesis or anti-angiogenic/therapeutic targets?

Author Response

Thanks for your valuable comments, our response is as follows

Comment 1:  In 2023, a very similar study was published by Lei Zhou and published in the BMC Gastroenterol. In my opinion, an update of the literature should take into consideration this study, adding what has been published in the last two years.

Response: In the discussion paragraph 1, we updated the two literatures about CXCL14 study in colorectal cancer.

Comment 2: Have the authors cross-checked their results about CXCL14  in the available Colon cancer data in cBioPortal.

Response: We studied the correlation of CXCL14 mRNA and overall survival time with the dataset of Colon Cancer (Sidra-LUMC AC-ICAM, Nat Med 2023)that available on cbioportal database, and found the similar positive correlation.

Comment 3: The authors mentioned in the methodology section about HCT15 cell lines, but not shown in the results section, can authors discuss the results of this HCT15 cell lines?

Response: The result of HCT15 has the similar trend with HCT116, so we just showed the result of HCT116. Here, we deleted the “HCT15”from the Method section.

Comment 4:  Could the authors add more in the conclusion about the role of CXCL14 in the progression of colon cancer angiogenesis or anti-angiogenic/therapeutic targets?

Response: We have added the discussion about the CXCL14 angiogenesis at the fourth paragraph in the Discussion section.

Reviewer 2 Report

Comments and Suggestions for Authors

This is a very interesting study in which the authors investigated in silico and in vitro how the methylation status of CXCL14 changes during CRC progression according to the Vogelstein model and in mice with inflammation and colitis-associated cancer (CAC). It was found that CRC progression is partly a CXCL14 methylation-dependent process, and this was confirmed in a colon adenocarcinoma cell line by methylation and demethylation experiments.
The basic idea, design, and execution of the article are very good.
However, I have a few suggestions:
- 15 IBD samples were analyzed in the analysis of human samples. I would recommend removing these samples from Figure 1 and Table 1, as the other samples in the work of Galamb et al. were not colitis-associated cancers but sporadic cancers, according to the Vogelstein model.
However, if they want to keep the human IBD samples, they should compare this group with human colitis-associated cancer samples, as this would fulfill the Itkovitz model, i.e., the CAC development pathway.
- if the human IBD data are retained, the demographic and disease characteristics of IBD patients should be published.
After the above corrections, I recommend accepting the article for publication.

Comments on the Quality of English Language

Minor polishing, stylistic, and grammatical editing are necessary. 

Author Response

Thanks for your valuable comments, our response is as follows

- 15 IBD samples were analyzed in the analysis of human samples. I would recommend removing these samples from Figure 1 and Table 1, as the other samples in the work of Galamb et al. were not colitis-associated cancers but sporadic cancers, according to the Vogelstein model.
However, if they want to keep the human IBD samples, they should compare this group with human colitis-associated cancer samples, as this would fulfill the Itkovitz model, i.e., the CAC development pathway.
- if the human IBD data are retained, the demographic and disease characteristics of IBD patients should be published.
After the above corrections, I recommend accepting the article for publication.

Response: We have deleted the IBD data from table 1 and figure 1.

Reviewer 3 Report

Comments and Suggestions for Authors

The CXCL14 gene represents the chemokine family that can synergistically promote the physiological activity of several other chemokines. This manuscript studied the changes in CXCL14 expression from normal tissue to colitis by analyzing multiple datasets. Moreover, experiments on 293T, HCT15, and HT29 cell lines were done. The authors proved the dynamic expression of this gene. Notably, CXCL14 expression is up-regulated in early cancerous tissues and undergoes silencing in metastatic tumors via its promoter hypermethylation.

Low expression of CXCL14 is correlated with shorter survival time of patients with colorectal cancer. This data can be essential from the predictive and therapeutic points of view.

I have minor issues concerning this manuscript:

1. Please add the source of cell lines used in the experiments. 

2. Is the 70% humidity a proper information, while typically I know the higher percentage?

3. Please unify the text formatting and always use the full name of the Figure, e.g., Figure 3A instead of Fig.3A.

Author Response

Thanks for you valuable comments, our responses are as follow.

  1. Please add the source of cell lines used in the experiments. 

Response: We have added the cell lines source.

  1. Is the 70% humidity a proper information, while typically I know the higher percentage?

Response: We checked the protocols of cell culture, in some labs the 95% humidity was used, but we measured our cell incubator, the humidity is 70%.  In the future experiment, we will pay more attention on the humidity.

  1. Please unify the text formatting and always use the full name of the Figure, e.g., Figure 3A instead of Fig.3A.

Response: We have corrected all names of Fig. to Figure.